# Development of a 1D/2D Urban Flood Model Using the Open-Source Models SWMM and MOHID Land

João Barreiro [1,*], Flávio Santos [2], Filipa Ferreira [1], Ramiro Neves [2] and José S. Matos [1]

1   CERIS (Civil Engineering Research and Innovation for Sustainability), Instituto Superior Técnico, Universidade de Lisboa, Av. Rovisco Pais, 1049-001 Lisbon, Portugal
2   MARETEC (Marine, Environment and Technology Center), Instituto Superior Técnico, Universidade de Lisboa, Av. Rovisco Pais, 1049-001 Lisbon, Portugal
*   Correspondence: joao.barreiro@tecnico.ulisboa.pt

**Abstract:** Urban pluvial floods are the outcome of the incapacity of drainage systems to convey the runoff generated by intense rainfall events. Cities have been struggling to control such hazards due to several pressures, such as urbanization increase, more frequent experiences of extreme rainfall events, and increases in tide levels. Such pressures demand the study of adaptation strategies, which conventional one-dimensional drainage models fall short of simulating. Thus, 1D/2D models have been emerging with the aim of allowing better integration of key processes for flood modeling, namely, runoff interception by stormwater inlet devices and manhole overflows. The current paper presents a 1D/2D urban flood model based on an offline coupling procedure between the 1D model SWMM and the 2D model MOHID Land. The SWMM/Land model is applied to a synthetic street case study and to a real case study in downtown Albufeira, Portugal. The results obtained for the real case study are coherent with local observations of past flooding events, and the model shows potential for better decision-making regarding urban flood risk management.

**Keywords:** 1D/2D flood model; MOHID; SWMM; urban drainage; urban flooding

## 1. Introduction

Urban pluvial floods usually result from intense or extreme rainfall events, concentrated in time (typically few hours) and space, which may result in generated runoffs that are higher than the design capacity of the drainage systems [1,2]. IPCC reported that people are increasingly experiencing unfamiliar precipitation patterns, including extreme precipitation events [3]. Climate change itself is not likely to change the nature of intense rainfalls, but will change their severity and frequency, and potentially their geographical range [4]. Moreover, the fast urban growth that has been ongoing since the last decades of the XXI century has led to profound changes in the pre-existing urban hydrologic cycle and has put existing infrastructures under stress. In 1950, 30% of the world's population lived in cities; in 2018, this fraction was 55%; and in 2050, it is expected to rise to 68% [5]. A clear and direct change promoted by urbanization processes is the decrease in the perviousness of the catchments, which leads to a decrease in the peak flow lag time and a decrease in flood peak duration, but with an increase in peak discharges, a reduction in groundwater recharge, and a consequent reduction in baseflow in urban streams [6].

Aging drainage infrastructures face not only demanding challenges (due to climate change, land use and demographic changes), requiring investments in new infrastructures, but also require proper rehabilitation to preserve their functionality from a long-term perspective. Decision-support tools regarding the prioritization of rehabilitation interventions benefit from condition assessment techniques and protocols, and from service-oriented approaches, supported by sewer system 1D/2D modeling, that minimize uncertainties and urban floods hazards [7,8].

Conventional drainage systems are designed with the purpose of getting rid of urban runoff and conveying it as fast as possible to an outfall, typically located at the nearest water body, usually at streams/rivers, lakes, or the sea [9]. The discharge condition is a critical factor in the performance of drainage systems, especially in coastal systems subjected to sea tides. The influence of tides on outfalls decreases the discharge capacity, promoting flow deceleration and upstream network surcharge. According to IPCC, the tides' influence on stormwater systems has been increasing due to climate change and it is certain that, in the near-term (2021–2041), continued and accelerating sea level rise will encroach on coastal settlements and infrastructure and, if trends in urbanization in exposed areas continue, this will exacerbate the impacts on urban services [3]. Additionally, meteorological-related events such as storm surges will pose higher pressures on stormwater discharges in coastal areas. This way, multiple factors interact, generating higher vulnerabilities to climate hazards and intensifying overall risk. Thus, future sea level rise combined with storm surge and heavy rainfall will increase combined flood risks [3].

Decision makers are in need of tools that can help them to decide on not only the best recovery actions after floods, but also how to tackle critical services/infrastructures under different disruptive scenarios [10]. In the case of urban flooding, modeling tools are critical to assessing drainage system performance and its influence on the city's overall response to rainfall events.

Typically, stormwater drainage models are composed of two steps: the first concerns the hydrologic processes, where rainfall is transformed into runoff as the outcome of a set of hydrological processes responsible for all the losses that rainwater undergoes when reaching the catchment surface [11]; the second concerns the runoff transport along the drainage network. In the second step, the drainage network is typically conceptualized as a set of nodes, corresponding to manholes, and connections/links between them, representing sewers or open channels reaches. The runoff generated in a given catchment is routed to an entry node, and is then conveyed along the drainage network by solving the one-dimensional (1D) Saint-Venant equations (SVE) [12].

However, the use of such models alone presents limitations concerning the interaction processes between runoff and the drainage systems, undermining its full potential for the study of urban floods. Firstly, simplifying the urban topography and assuming a set of sub-catchments leads to the neglect of several urban infrastructures that influence the runoff behavior, such as walls or terrain depressions. Secondly, assuming that the generated runoff in each sub-catchment is fully routed to the drainage system implies a 100% interception efficiency of the storm inlet devices, which tends not to be the reality. Thirdly, when the drainage system surcharges, outflows can result from manholes and the 1D models cannot deal with these flows' propagation at the surface, assuming that they are either loss from the system (Figure 1a) or ponded in a virtual volume above the flooding manhole, returning to the same node when possible (Figure 1b).

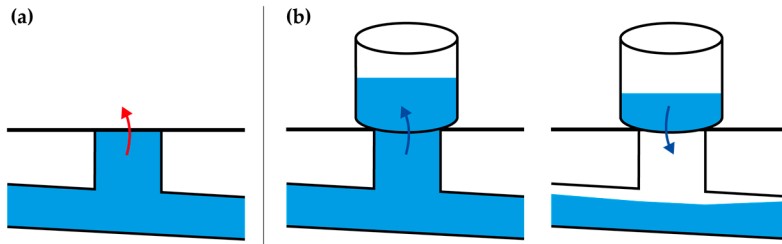

**Figure 1.** Graphical representation of 1D models dealing with manhole overflow: (**a**) lost from the system or (**b**) ponded in virtual storage volume.

In response to these issues, 1D/1D models were introduced, where both the runoff and drainage systems' flow are modeled by solving the 1D SVE (e.g., [13–15]). This way, streets are represented by open channels placed over the drainage network, and it is possible to consider flow exchanges between both manholes and inlet devices. Although there are

automatic GIS procedures for the delineation of the preferential courses of surface runoff, allowing faster assembly of 1D/1D models [16], the topography of cities is varied, and when runoff occurs in wide areas, it tends to assume a bidirectional behavior. Therefore, the 1D flow simplification should not be used to represent the runoff propagation [16,17].

As a consequence, 1D/2D dual drainage models arose, solving the 2D SVE for surface runoff and keeping a 1D approach for the flow in the drainage network. Recently, commercial software programs considering 1D/2D urban flood modeling have become available, such as Mike Urban by DHI [18], Infoworks ICM by Innovyze [19], and OpenFlows FLOOD by Bentley Systems [20]. These rely on licensed software, and the availability of open-source or freeware programs is still scarce in this domain [21].

Various 1D/2D modeling approaches can be found in the literature with different degrees of complexity regarding the physical processes involved. Some authors present simplifications of the 2D SVE, ignoring the inertial term [21–23]. Different approaches are also found regarding the inlets' interception capacity (e.g., capture flows up to a given maximum threshold [22] or inlet interception curves obtained from experimental studies [24,25]). Free weir and orifice equations are mostly found in the literature to compute the inlets' interception capacity and surcharged weir and orifice equations to define the manhole overflow [21,26–32] and inlet overflow [33], although the respective weir and orifice coefficients typically require calibration [26,34]. In other cases, inlet location is not considered and flow is assumed to be intercepted at the manhole locations [21,27,28,30,31,35].

The current paper presents a 1D/2D urban flood model based on an offline coupling procedure between the open-source models SWMM (1D) and MOHID Land (2D). Urban drainage infrastructures, namely manholes and inlet devices, are interface connectors between these models. Inlet devices are responsible for capturing runoff and convey it to manholes (2D to 1D), whilst manholes can return excessive flows to the surface when the sewer system is pressurized (1D to 2D). A new MOHID Land module was built, and no code modifications were made in SWMM. The offline coupling procedure allows both models to interchange data at the end of each run, through adequate timeseries files.

The current introductory chapter follows the presentation of the used models and the coupling procedure developed. Thereafter, the model is applied to two case studies: a synthetic street and a real case study in Albufeira, Portugal. At last, conclusions are presented concerning the model results and improvement opportunities.

## 2. Materials and Methods

### 2.1. Hydrologic and Hydraulic Models

#### 2.1.1. EPA-SWMM

The Storm Water Management Model (SWMM) is a hydrologic and hydraulic model that started to be developed by the United States Environment Protection Agency (EPA) in 1971 [36]. Since then, the model has undergone several upgrades, including the inclusion of the EXTRAN block in 1977, which allowed the 1D dynamic simulation of flow transport through channels and closed pipes; it was one of the most used blocks in various applications of SWMM. In 1988, version 4.0 was released, in Fortran-77 [37]. The most recent substantial update occurred in 2004, in version 5.0, with the rewriting of all the code in C language, allowing its compilation in dynamic-link library files (.dll) and enabling it to run on the command line via an executable file (.exe). Version 5.1, released in 2014, has undergone several updates, both to the computational methods and to the Graphical User Interface [38]. At last, in February 2022, version 5.2 was launched and its major new feature consists of a 1D/1D explicit approach, allowing users to define inlet devices that capture street runoff using the U.S. Federal Highway Administration's HEC-22 methodology [39]. Apart from sewers and manholes, SWMM allows us to consider a wide range of drainage infrastructures such as storage/treatment facilities, pumps, and flow regulators.

The basic unit of the SWMM rainfall-runoff model is a catchment. SWMM uses a non-linear reservoir model to estimate runoff by conceptualizing a catchment as a rectangular surface with uniform slope and width. From the conservation of mass, the net change

of the water depth per unit of time is the difference between inflows (rainfall rate) and outflows (infiltration, evaporation, and runoff rates) over the catchment [40].

### 2.1.2. MOHID Land

MOHID Land, developed by the MARETEC (Marine and Environmental Technology Research Center) at the Instituto Superior Técnico of the University of Lisbon, is a hydrologic–hydraulic integrated model with four compartments: the atmosphere, porous media, surface land, and river drainage network (Figure 2). It is part of a broader model, MOHID, an open-source model written in Fortran, which also includes MOHID Water, a three-dimensional numerical program to simulate surface water bodies.

In MOHID Land, water moves through the media based on solving the complete SVE (2D in surface runoff and 1D in river networks), also allowing for kinematic wave and diffusion wave approximations. The atmosphere is not explicitly simulated but provides data necessary to impose boundary conditions (precipitation, solar radiation, wind, etc.) on the remaining compartments. The model is based on finite volumes arranged in a structured grid. Surface land is described by a 2D horizontal grid and the porous media by a 3D domain that includes the same horizontal grid as the surface, complemented by a vertical grid with layers of varying thickness. Infiltration can be calculated by different models, namely, the Curve Number model developed by the Soil Conservation Service, the Green-Ampt model, and the Richards Equation, which is also used to model the movement of water along the soil porous media. The river drainage network is a 1D domain defined from the digital elevation model (DEM), with reaches connecting the centers of the surface cells. Fluxes are calculated over the faces of the finite volumes, and state variables are calculated at the center to ensure the conservation of transported properties. The model uses an explicit algorithm with a variable time step [41,42].

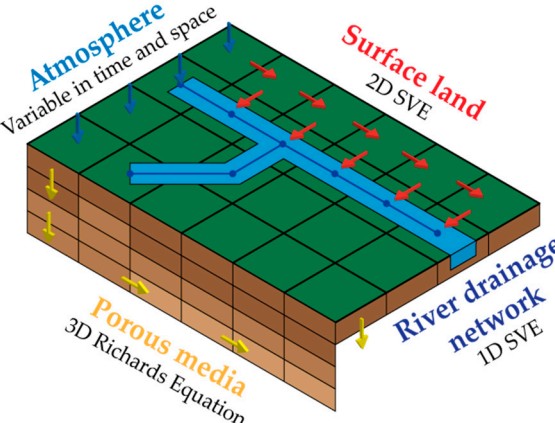

**Figure 2.** MOHID Land compartments adapted from [42].

The basic unit of the MOHID Land rainfall-runoff model is a cell of the 2D horizontal grid representing the water elevation, i.e., the sum of the surface elevation and the water column at each cell. Thus, likewise SWMM, from the mass balance, the net change in the water depth per unit of time is the difference between the inflows (rainfall rate) and outflows (infiltration and evaporation rates) over the cell, with the difference of adding/subtracting the water fluxes from/to the neighboring cells.

### 2.2. SWMM/Land Coupling Methodology

### 2.2.1. Coupling Rationale

The interest in coupling MOHID Land with SWMM is due to the possibility of better reflecting real flooding behavior due to interactions between the runoff and the flow in the drainage network, namely the runoff capture by inlet devices and the flow propagation at

the surface when overflow through manholes occurs. Considering this, the fundamentals of the coupling rationale are as follows (Figure 3):

1. Stormwater inlets capture surface runoff (on MOHID Land) and route it to the urban drainage network. The captured water will decrease the water level at the surface, i.e., the water column in the corresponding cell of the 2D surface grid. In abnormal conditions, flooding occurs if the stormwater inlets are insufficient in number and/or have insufficient capacity, leading to water accumulation at the surface.
2. Water captured by the stormwater inlets is conveyed by the urban drainage network and discharged at the outfall (on SWMM), often a river or sea.
3. When the captured flow surpasses the carrying capacity of the urban drainage infrastructures, such that the water depth inside the network reaches the surface level, part of the captured water can return to the surface through manhole overflow. If a given manhole is overflowing, the intake capacity of the connected inlet devices is set to zero.

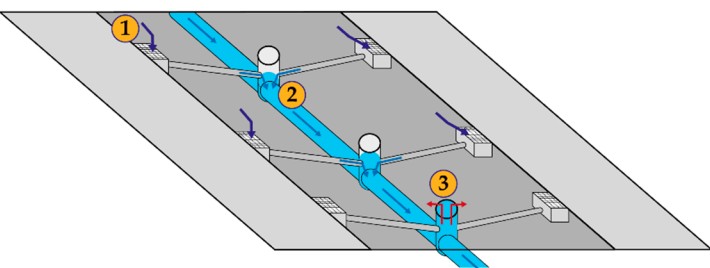

**Figure 3.** Fundamentals of the SWMM/Land coupling rationale.

### 2.2.2. Data Requirements

The main information required to assemble a MOHID Land domain in a typical application in an urban environment consists of georeferenced data that enable the building of different 2D horizontal grids, to be used in the computation of the involved hydrologic/hydraulic processes (Table 1).

**Table 1.** Main data required, typical formats, and respective uses in MOHID Land.

| Data Required | Typical Formats | Relevant Grid | Main Process |
|---|---|---|---|
| Contour lines/DEM | Vectorial/raster | Surface grid | Runoff |
| Buildings and urban obstacles | Vectorial | Surface grid | Runoff |
| Land use | Vectorial/raster | Manning's coefficient grid | Runoff |
| Imperviousness factor | Raster | Imperviousness grid | Infiltration |
| Soil properties | Vectorial/raster | Soil properties [1] | Infiltration |

[1] One grid by soil propriety. Required soil properties depend on infiltration method chosen.

Regarding the SWMM component, the key data needed are the infrastructures' registers with their basic properties and characteristics, also georeferenced (Table 2).

**Table 2.** Main data required for SWMM implementation.

| Infrastructure | Main Data Required |
|---|---|
| Manholes | Invert elevation; depth from invert to ground |
| Sewer/channel | Cross-section shape; length; roughness |
| Storm tank | Invert elevation; depth of the storage unit; storage curve (surface area as function of water depth) |
| Pump | Pump type and curve; startup and shutoff depths |
| Outfall | Invert elevation; discharge boundary condition |

The rationale for the coupling between MOHID Land and SWMM requires one additional layer of data able to capture runoff from the surface, that is, to model inlet devices (Figure 4).

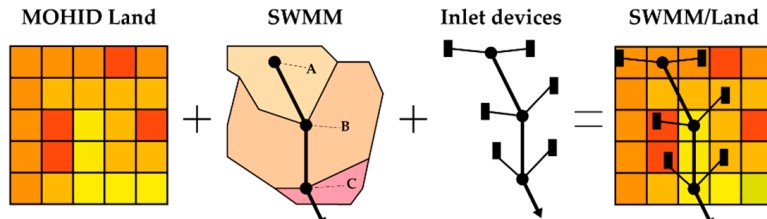

**Figure 4.** Data layers considered in the rationale of the coupling methodology between MOHID Land and SWMM.

As mentioned before, SWMM assumes that all the runoff generated by a given subcatchment is conveyed to its respective outlet node, assuming that inlet devices have the capacity to fully capture runoff, which is not true for most real situations. In this sense, stormwater inlets are not directly considered in typical SWMM 1D applications (although if calibration data exist, the efficiency of the inlet devices is blended in data with other processes). Within the current coupling, the need for additional data regarding the inlet devices, namely their location and geometry, will result in a better representation of the real processes, which is the main goal of the model.

2.2.3. Coupling Procedure

In practice, the current coupling follows an offline procedure, i.e., models interact by changing data through timeseries by the end of the run of each model. MOHID Land is responsible for generating runoff due to rainfall, and when runoff reaches an inlet device, it is partially removed from the cell and the captured flow is written on a timeseries. SWMM is responsible for conveying the captured flows along the drainage network and for writing timeseries regarding outfall discharges and manhole overflow (if existent). In the end of the MOHID Land run, there will be two main outputs in the form of timeseries that will be used on SWMM: the captured flow by the manhole, i.e., the sum of the flows captured by all the inlet devices linked to that manhole, and the surface water level at each manhole location. The first will be used as flow input at each SWMM node (manhole) and the second as a boundary condition in the case of manhole overflow. The manhole overflow is conditioned by the water head at the surface as the overflow value is computed as a function of the gradient between the water head in the SWMM node and the water head at the surface. The run of SWMM produces a timeseries containing the flows leaving the model through outfall discharges and due to manhole overflow.

Thus, the run of the urban flood model is composed of runs of cycles: a first run of MOHID Land followed by a run of SWMM (Figure 5). Since the coupling is offline, the interchange of flows is always dependent on the conditions of the previous run. A procedure to analyze the need for successive runs is also proposed, the S/L-OCA (SWMM/Land Operational Convergence Analyst). This procedure allows us to verify if there is a need to proceed with another simulation cycle by analyzing two criteria:

1. Manhole overflow occurrence: if there is any manhole overflow during the simulation time, another simulation cycle is required to model such inflows at the surface.
2. Water depth convergence between cycles: from the second cycle onwards, the water depth results in a set of user-defined 2D probe cells that are compared, and a new simulation cycle is required if the convergence is smaller than a given user-defined threshold.

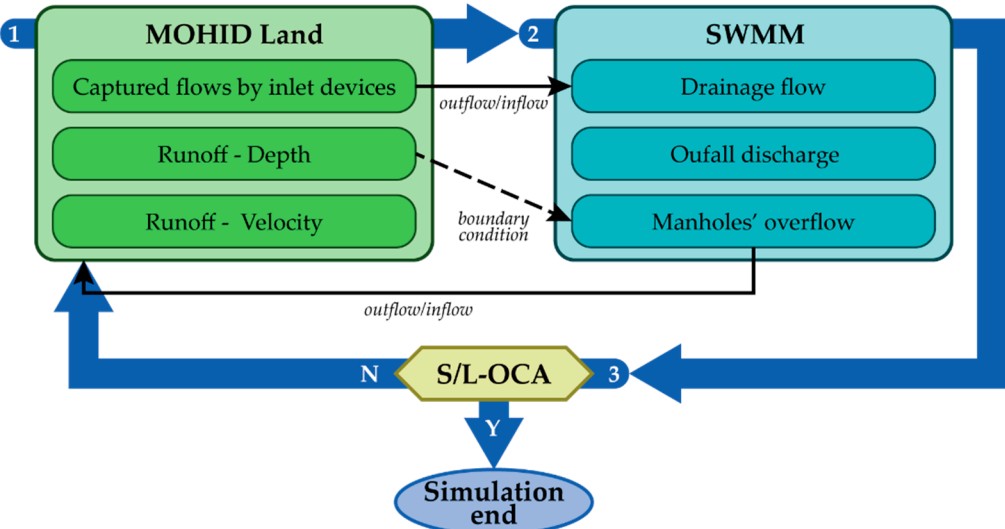

**Figure 5.** SWMM/Land coupling steps and simulation cycles.

The presented coupling procedure also allows for the rainfall-runoff process to be simulated by SWMM, starting with step 2 of Figure 5. This way, SWMM built-in functionalities that operate on SWMM catchments can still be considered, such as Low-Impact Developments. If such an option is chosen, an interaction between the flow in the drainage network and the surface only exists if manholes overflow.

The current implementation required the creation of a new module for MOHID Land and specific adaptations/settings for the use of SWMM, the latter with no change in the code. The new module for MOHID Land allows the model to read urban drainage infrastructures, namely, manholes and inlet devices, fed to the model through ASCII files.

Thus, MOHID Land is capable of capturing runoff form each cell containing at least one inlet device. For the current implementation, inlet captured flow is mediated by a weir equation (Equation (1)) [21,26].

$$Q_{2D/1D} = c_w \times L \times h_{2D}^{3/2} \times \sqrt{g} \tag{1}$$

where $Q_{2D/1D}$ is the inlet captured flow (m$^3$/s), $c_w$ is the weir coefficient (assumed as 0.2), $L$ is the length of the inlet, $g$ is the acceleration of gravity, and $h_{2D}$ is the water depth at the MOHID Land cell.

Concerning SWMM, to each node corresponding to a manhole, an outfall link was added through an outlet link type. This way, the runoff head is considered as the outfall boundary condition and the link mediates the manhole outflow rate through an orifice equation (Equation (2)) [21,27–29].

$$Q_{1D/2D} = c_o \times A_{mh} \times \sqrt{2g(z_{1D} - z_{2D})} \tag{2}$$

where $Q_{1D/2D}$ is the manhole overflow rate (m$^3$/s), $c_o$ is the orifice coefficient (assumed as 0.5), $A_{mh}$ is the area of the manhole, $g$ is the acceleration of gravity, and $(z_{1D} - z_{2D})$ is the difference between the water head at the SWMM manhole, $z_{1D}$, and the respective MOHID Land cell water surface elevation, $z_{2D}$.

## 3. Application to Case Study and Results

### 3.1. Synthetic Case Study: Simplified Street

This case study is suggested by Sañudo et al. [32] to allow the verification of the coupling procedures. It consists of modeling a synthetic domain with street elements as described in Table 3. The drainage system consists of four manholes (M1, M2, M3, and M4) and one outfall (O1), linked with sewers with a 1% slope, and eight inlets connected to the

nearest manhole. The sewer along the road has an inner diameter of 500 mm, whilst the sewer connecting M4 to M2 has an inner diameter of 300 mm. The domain elements and the 2D elevation grid, composed of a $0.5 \times 0.5$ m structured mesh of 2780 cells, are shown in Figure 6.

**Table 3.** Street elements of the synthetic domain [32].

| Element | Dimension (m) | Slopes (%) | Manning Coefficient (s/m$^{1/3}$) |
|---|---|---|---|
| Road | L = 40, W = 7 | $S_T = 2$, $S_L = 1$ | 0.016 |
| Sidewalk | L = 40, W = 2 | $S_T = 1$, $S_L = 1$ | 0.016 |
| Green area | L = 14.5, W = 10 | $S_T = 1$, $S_L = 1$ | 0.032 |
| Buildings [1] | L = 40, W = 10 L = 14.5, W = 10 | - | - |

Legend: L: length, W: width, $S_T$: transversal slope, $S_L$: longitudinal slope. [1] Buildings are not considered for the 2D mesh, being directly linked to the drainage system.

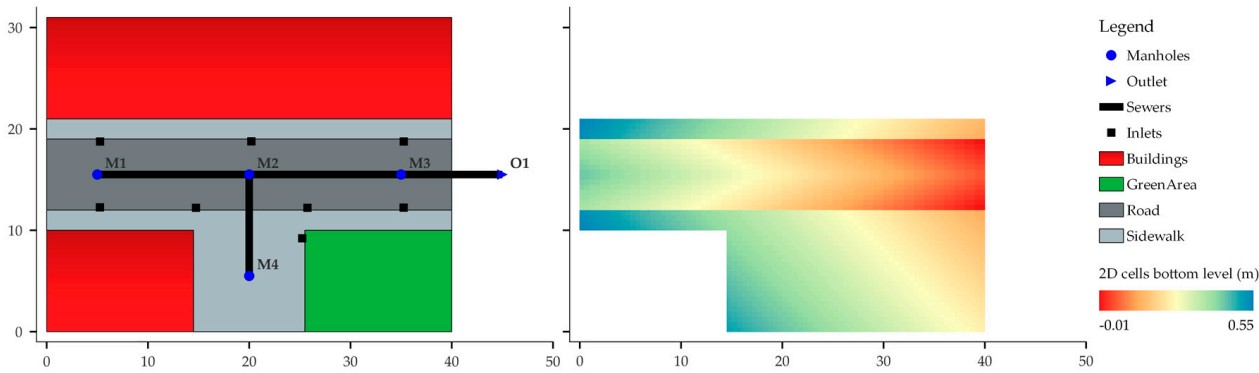

**Figure 6.** Synthetic case study: SWMM/Land domain.

The suggested boundary conditions are set to force the system surcharge and simulate manhole overflow conditions: the manholes M1 and M4 are forced with input hydrographs, the outfall is forced with a conditioning boundary level, and a constant rainfall intensity of 80 mm/hour is imposed (Figure 7a). Figure 7b presents the results regarding inflows and outflows at each manhole, i.e., runoff captured by the inlet devices and manhole overflow. Figure 8 presents the water head at the SWMM nodes and at the respective 2D cell.

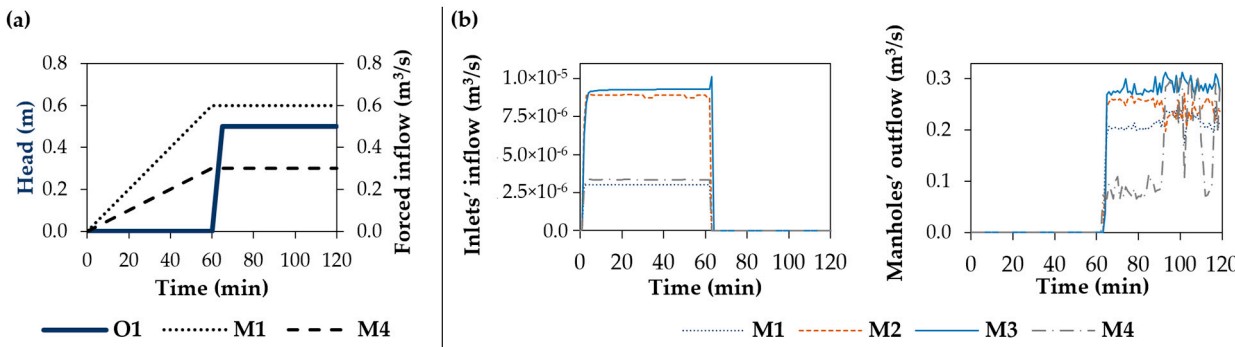

**Figure 7.** Synthetic case study: (**a**) imposed boundary conditions; (**b**) captured runoff by inlet devices and manhole overflow.

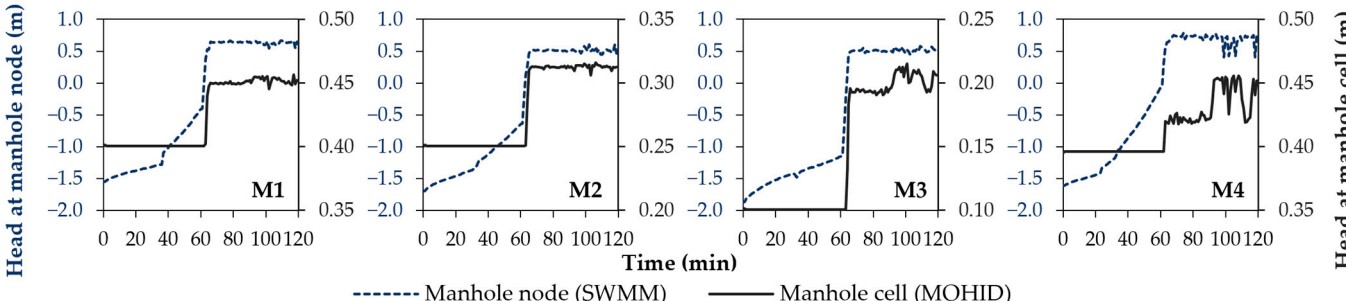

**Figure 8.** Synthetic case study: resulting water head at manholes (SWMM nodes) and at respective 2D cell (MOHID).

The resulting behavior is consistent with the coupling rationale and procedure presented. As expected, the head in manholes increases gradually up to the 60 min due to the imposed hydrographs and the inflow from inlets and buildings. When reaching 60 min, due to the boundary condition at the outfall, the system rapidly surcharges, manholes start to overflow and, consequently, inflow from the inlets stops. As stated, the outflows are mediated by the gradient between the head at the nodes and the respective surface cell. The higher variability of manhole M4's outflow and respective water head is due to a local deceleration of the flow on its surroundings, namely on the nearest inlet, caused by the green area and its associated Manning coefficient.

Figure 9 represents the maximum water depth and velocity modulus at maximum water depth, in the cells of the 2D domain. The manholes' locations stand out clearly in both results due to the manhole overflow effect. It is observed that the flow spreads accordingly with the road slope, downstream and towards the curb direction, with increasing velocity along the road. There are three interesting observations regarding the velocities at maximum water depth: firstly, there is a local effect of flow acceleration around manhole locations due to the overflows; secondly, there is a deceleration effect upstream of the manholes due to the higher water depths in the cells representing manholes; thirdly, the higher Manning coefficient in the green area results in lower velocities, as expected, and due to such lower velocities, the water is captured by the nearest inlet.

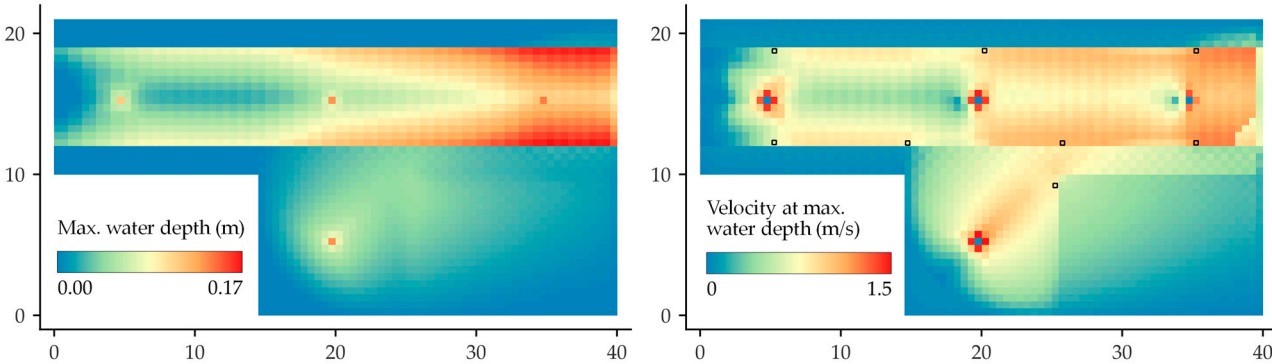

**Figure 9.** Synthetic case study: resulting maximum water depth (**left**) and velocity modulus at maximum water depth (**right**).

The presented results relate to the third run of MOHID Land, meaning that stability was reached at this point. The MOHID Land simulation times were similar between the three runs (18 m 43 s, 19 m 04 s, and 19 m 23 s), with an average ratio of simulation time over real time of 16%. The simulation times of the three SWMM runs were less than one second.

*3.2. Real Case Study: Downtown Albufeira*

3.2.1. Real Case Study Description

Albufeira is a coastal town located in Algarve, in the south of Portugal, bathed by the Atlantic Ocean. Downtown Albufeira is located at the final section of Albufeira watershed. The watershed has an area of 26.6 km$^2$ and a 4% average slope. The Albufeira creek develops naturally until it reaches the urban area, where it is piped along a stormwater drainage tunnel. This tunnel has an initial rectangular section of $3.0 \times 2.5$ m and discharges into Pescadores' beach; it is recurrently silted up with sand, which limits its discharge capacity. A sea outfall with a diameter of about 1 m is located at the outfall of this tunnel to convey polluted overflows from the wastewater system into the sea, avoiding potential contamination of coastal waters by non-rainwater that may flow into the tunnel. However, it is silted up frequently, thus having deficient performance. A smaller part of the downtown area watershed of about 40 ha is drained into a minor tunnel with an initial sectional of three barrels of $0.8 \times 1.2$ m, which suffers some changes along its length, namely cross-section reductions. This tunnel discharges into the sea through a moving bed pontoon, approximately 4 m wide, which also presents discharge limitations due to sand accumulation, as a result of non-self-cleansing velocities and tidal effects [43]. Figure 10 presents the Albufeira watershed and the main urban drainage infrastructures described above, along with the urban altimetry. The concave altimetry of the urban area is a major hazard for the occurrence of urban floods.

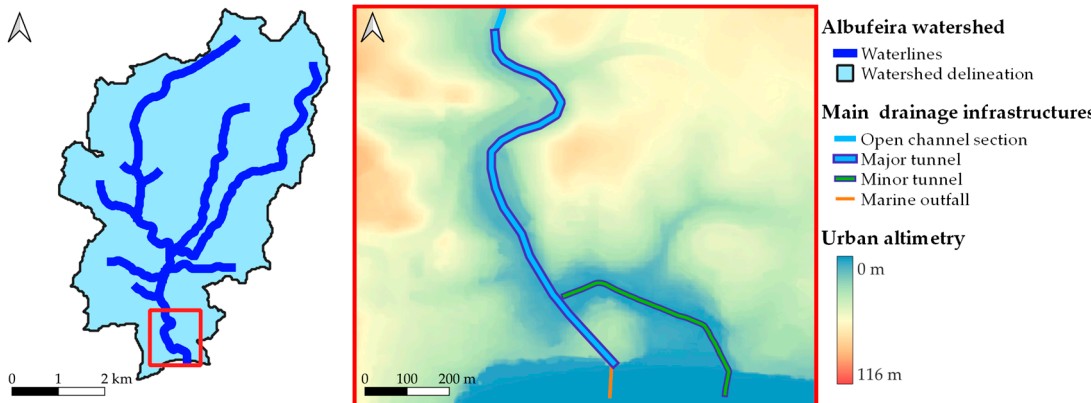

**Figure 10.** Albufeira watershed and main urban drainage infrastructures and altimetry.

Figure 11 presents photographs of flooding events that occurred in September 2008 and November 2015. The top left and top right images were taken in different events, at the same location. Although the study area has no drainage monitoring data, comparison of the rainfall data registered by rain gauges located in its surroundings with respective rainfall probability curves induces an estimated return period of about 10 and 100 years for these events, respectively [43].

The 2D regular mesh used to assemble the urban MOHID Land model is composed of 52,030 cells with a spatial resolution of 5 m. Altimetry data was obtained from a LiDAR DTM of a 1 m spatial resolution, and the surface imperviousness factor was derived from Copernicus Land monitoring services [44]. Vectorial data regarding green and constructed areas and building delineation were delivered by the municipality. The first data set was used to define different Manning coefficients (0.015 for streets, 0.030 for urban green areas, 0.065 for upstream flood plain, and 0.023 for others [45]), whilst the latter was used to raise by 15 m the elevation of cells that have at least 80% of their area covered by buildings. Infiltration was calculated using the Green-Ampt model and the soil parameters were selected according to the topsoil USDA classification database for Europe [46] and reference values [47]. The SWMM model is composed of 1168 inlets, 403 manholes, 11.9 km of sewers, and three outfalls. Such data were obtained from the drainage infrastructures

register from the municipality and complemented by satellite imagery. Figure 12 depicts the SWMM/Land domain and considered drainage infrastructures.

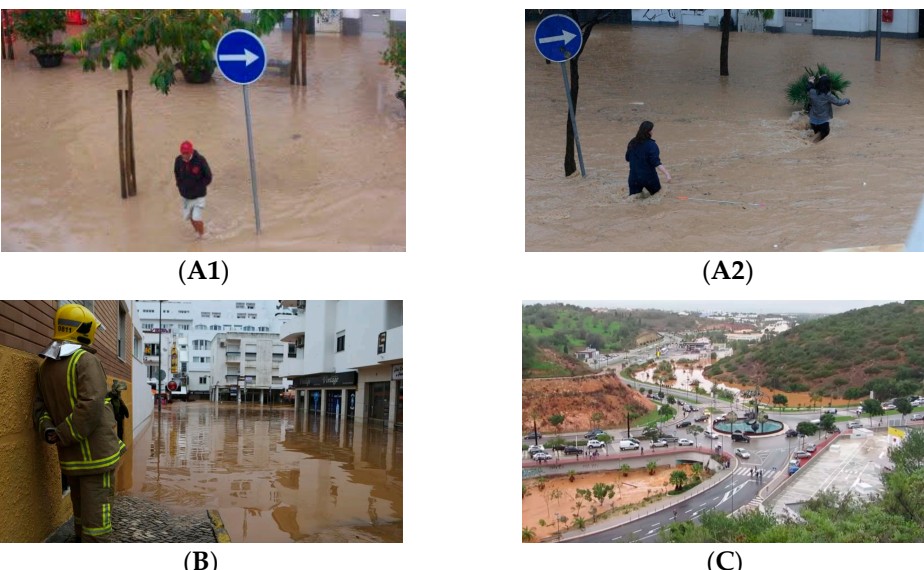

**Figure 11.** Consequences of the rainfall event that occurred in September 2008 (**A1**) and November 2015 (**A2,B,C**).

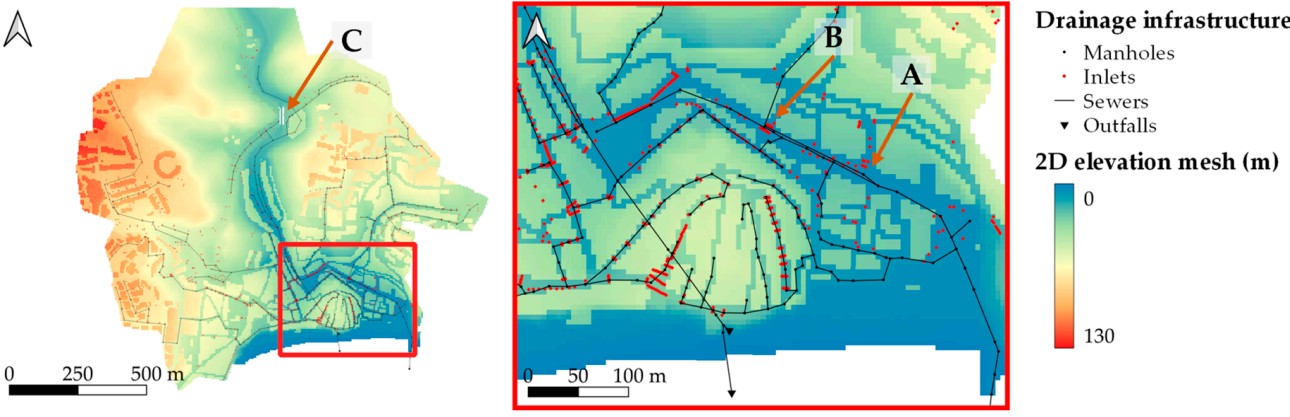

**Figure 12.** Albufeira case study: full SWMM/Land domain (**left**) and detailed view (**right**) with the location of the photos (A, B, and C) presented in Figure 11.

The described model was submitted to a project rainfall hyetograph fitted to the precipitation regimes in Portugal, with a total duration of 4 h and a centered rainfall peak of 1 h [48]. The rainfall intensities were estimated using intensity–duration–frequency (IDF) curves estimated to Faro [49], the nearest location with such statistical treatment of rainfall. Two return periods were considered: 2 and 10 years. Additionally, outfalls were exposed to tide levels, with the maximum high tide reaching two meters and coinciding with the critical period of rainfall (Figure 13, left). MOHID Land was also used on the upstream watershed area to obtain the Albufeira's creek hydrograph as inflow into the major drainage tunnel (Figure 13, right). The simulations were run for a complete day.

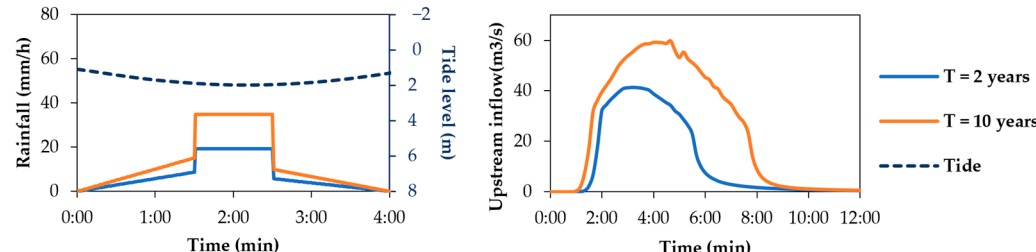

**Figure 13.** Albufeira case study: rainfall intensities by return period and tide level (**left**) and upstream inflow generated by the rainfalls with different return periods (**right**).

### 3.2.2. Real Case Study Results

Figure 14 presents the results obtained regarding the maximum water depth at each cell. As expected, higher maximum water depths are observed for the 10-year return period rainfall. The 2-year return period rainfall already results in maximum water depths of around 10–20 cm in some areas of the downtown area, owing to its concave orography. In the case of the 10-year return period rainfall, in addition to the aggravation in the area and the depth of the maximum water height reached in the lower part of the city, a retention effect is also visible in the upper part of the domain. This retention occurs in a green park located in the northern area of the city, where the entrance of the major drainage tunnel is located. The effect of retention occurs since the tunnel's entrance is incapable of conveying the total flow generated by Albufeira's creek, leading to the accumulation of water volumes in this area.

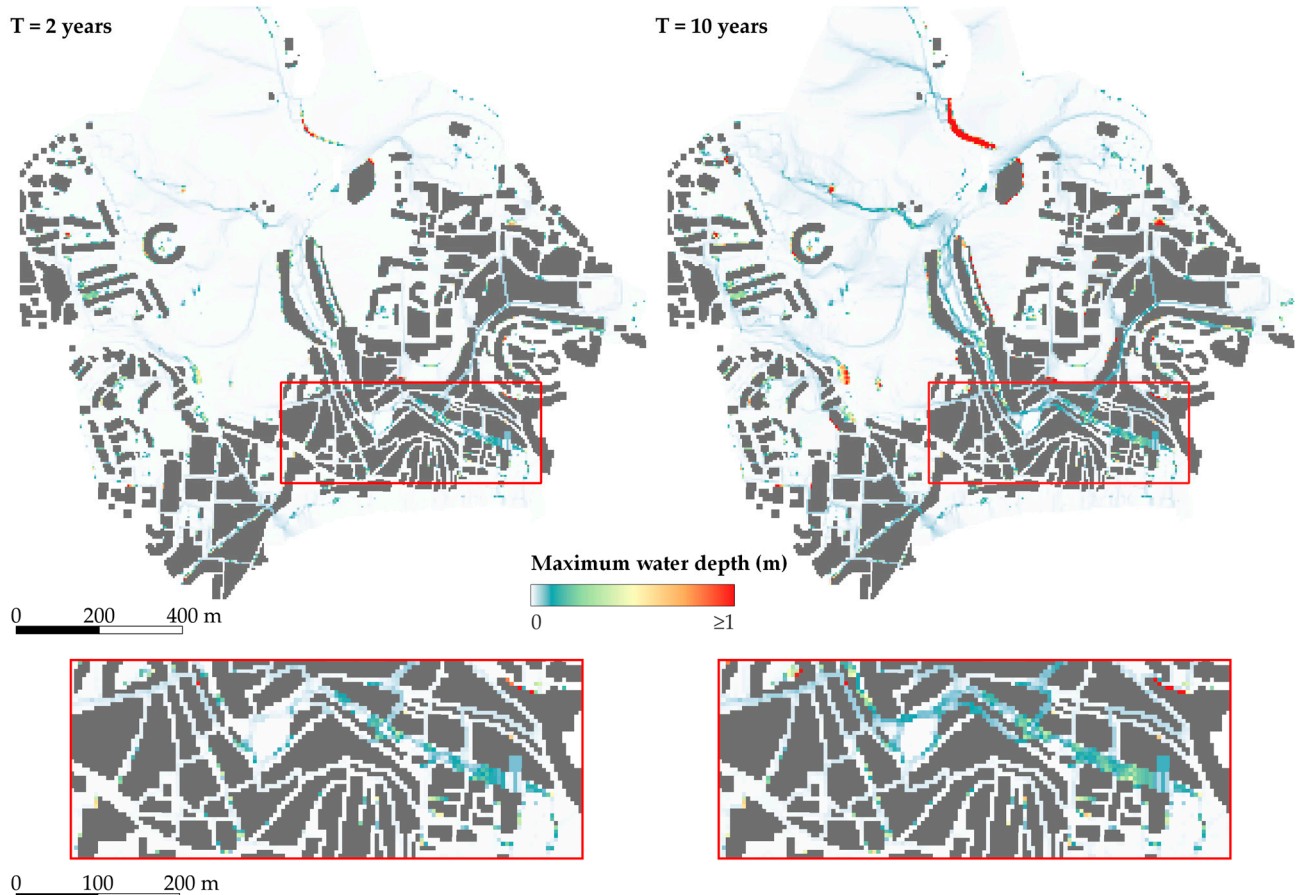

**Figure 14.** Albufeira case study: maximum water depth distributions for the 2- and 10-year return period rainfall.

The total inlet inflow and the percentage of the domain area affected by maximum water depth ranges are presented in Figure 15. The behavior of the inlet inflow evidences higher inflows in the case of the 10-year return period rainfall, as expected due to the higher generated runoff. Additionally, there is a slowly decreasing plateau reached after the rainfall event that is caused by the water volumes accumulating at sag cells that have inlet devices. Since these cells have lower elevations than their neighbor cells, the neighbor water volumes are attracted to these cells due to the inlet water abstraction.

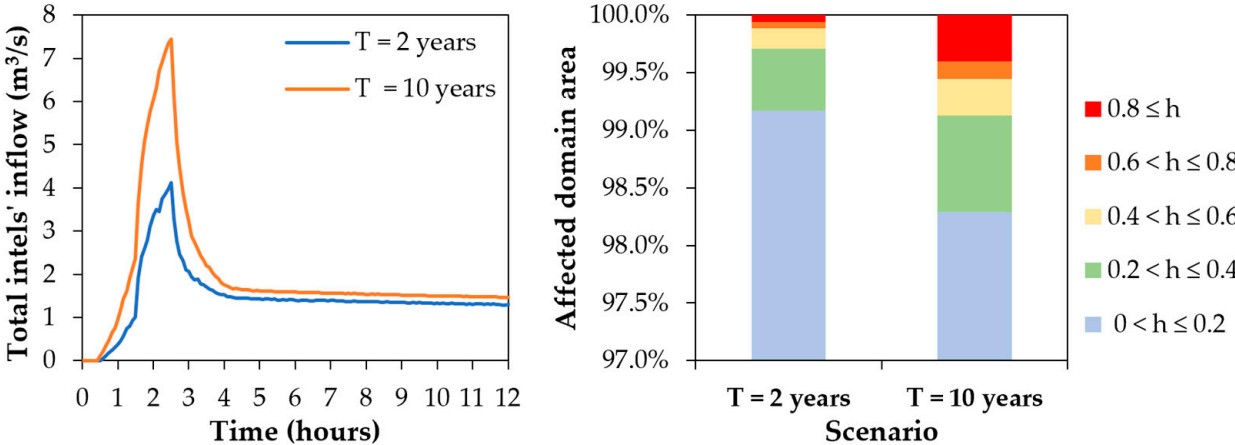

**Figure 15.** Albufeira case study: total inlet inflow (**left**) and percentage of the domain area affected by maximum water depth ranges for the 2- and 10-year return period rainfalls (**right**).

Figure 16 presents the inflow at three manholes; for example: manhole 1 is located on a narrow street in a location with higher elevation and has one inlet device assigned; manhole 2 is located near the downtown area and has three inlet devices assigned; and manhole 3 is in a critical downtown area, with four inlets assigned. The inflow behavior is coherent with the rainfall pattern as most of it is captured up to the fourth hour of the simulation. As manhole 3 is in downtown, where water volumes accumulate, runoff is captured over a longer period of time, gradually decreasing.

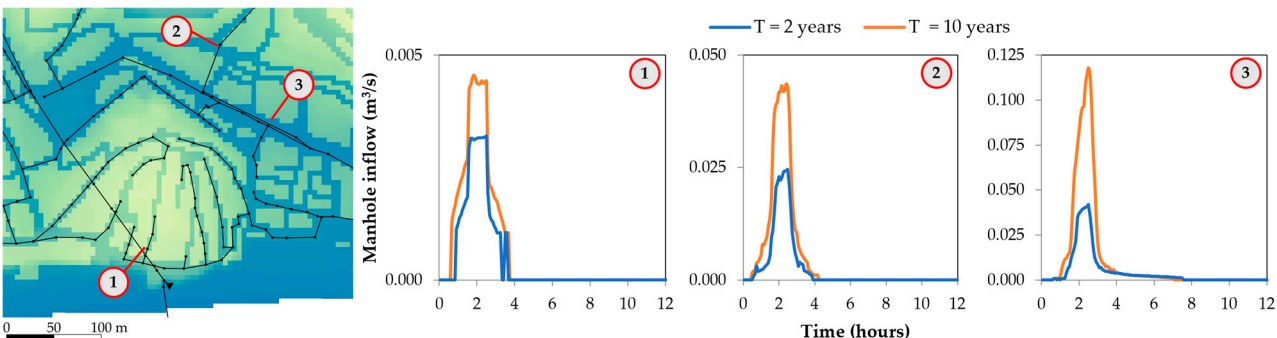

**Figure 16.** Albufeira case study: example of inflows in three manholes at locations 1, 2, and 3.

The hydraulic profile at maximum water depth of the major and minor drainage tunnels for the 10-year return period rainfall are presented in Figure 17. These profiles evidence that, although sewers surcharge, the water level does not reach the surface level for the 10-year return period. Nonetheless, the available spare capacity up to the surface level is reduced in both infrastructures.

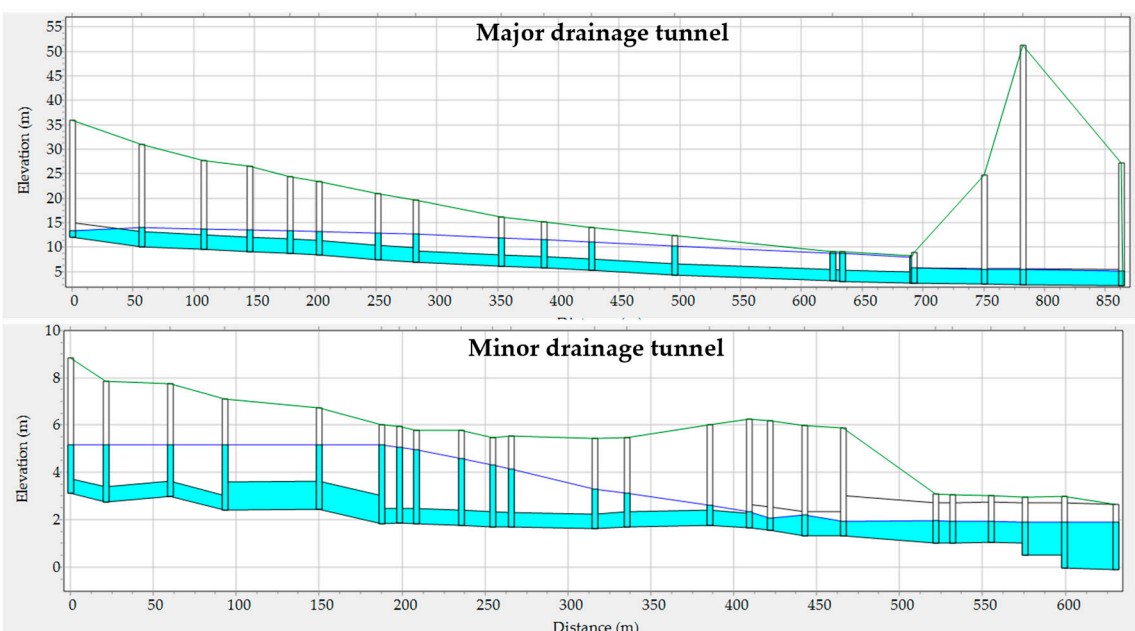

**Figure 17.** Albufeira case study: hydraulic profile at maximum water depth of the major and minor drainage tunnels for the 10-year return period rainfall (**top** and **bottom**, respectively).

Such results reveal that water accumulation at the surface is strongly related with the inlet devices' inefficiency along the urban catchments and highlight the relevance of the 1D/2D simulation to model such processes. The inlet devices for events with higher return periods have dual behavior: on one hand, with higher generated runoff, higher inflows are expected to be captured, contributing to the surcharge of the sewers and potentializing manhole overflow in downstream locations. On the other hand, as the global efficiency of the inlet devices decreases with higher runoffs, the runoff at the surface keeps increasing along the urban catchments, inevitably accumulating at low points and sag areas.

As no manholes overflowed, no second iteration was needed, and the presented results are related to the first iteration of SWMM/Land. The MOHID Land simulation times were 1 h 08 and 1 h 58 for the 2- and 10-year return period rainfalls, respectively. These running times mean a ratio of simulation time over real simulated time of 4% and 8%, respectively. The simulation times of the SWMM runs were 22 and 21 s, respectively.

## 4. Conclusions

The coupling between the SWMM and MOHID Land models lies in the assumption that regular 1D simulation models fall short of simulating runoff inflow limitations due to inlet performance constraints and manhole overflow processes. The proposed coupling procedure between two open-source models allows us to consider such processes along the simulation time. Storm inlet devices and manholes are considered contact elements between the two models: the first allows the interception of surface runoff and the second is responsible for the return of excessive inflows to the surface, through manhole overflow.

The SWMM/Land coupling procedure lies in an offline methodology where each model is run sequentially and they exchange information through resulting timeseries. The theoretical principle is that the more simulation cycles are run, the more accurate the results. For such purposes, a results analysis (the S/L-OCA) that investigates the need to repeat the simulation cycle is also suggested, based on the existence of manhole overflow and on the convergence of the water depth results between the simulation cycles. Although online coupling could be preferable, the methodology herein developed allows us to couple other 1D models with MOHID Land with ease, as the only change required is the conversion of the resulting timeseries to formats readable by each model. Moreover, an online procedure would require more profound changes in computational coding for both models.

Two case studies were considered: a synthetic case study representing a simplified street with a 2D grid made of 2780 cells of 0.5 × 0.5 m, and a real case study in Albufeira, Portugal, with a 2D grid composed of 52,030 with a spatial resolution of 5 m. The results obtained for both case studies are coherent with the theoretical rationale of the coupling. When balancing the achieved simulation times and the results obtained, for the case studies presented, these are considered satisfactory and a step forward compared to conventional 1D urban drainage modeling. Naturally, if applied to larger simulation domains, i.e., with more 2D cells, the MOHID Land running times are expected to be larger and, if overflows occur, the overall iterative simulation times can be aggravated.

As for any model, calibration and validation are only possible in the presence of monitoring data, which we did not have. Nonetheless, considering that both models, SWMM and MOHID Land, present several parameters that allow their individual calibration, the calibration of the coupled model is also an assured possibility. The results obtained for the case of Albufeira also evidence the need to have an accurate elevation grid, especially in low-slope and -sag areas. It is important to emphasize that the simulated events are not real, as rainfall intensities are depicted by a project rainfall hyetograph. Nonetheless, despite the lack of urban drainage monitoring data, the obtained results regarding flooded areas and depths are aligned with past flooding events and critical areas identified by the municipality.

Additionally, and considering the complexity of the involved processes, improvements are always possible. Future works on this specific coupling procedure should address the possibility of defining other inlet interception and manhole overflow coefficients, expressions, or relationships based on tabular data, for example. In addition to manholes, overflow is a process that also occurs through inlets. As this process is not directly considered herein, and overflow is only considered in manholes, future developments might also consider such processes. Moreover, the current coupling procedure has the potential to be developed towards a direct coupling procedure, i.e., simultaneous running and data interchange between the models.

Finally, the use of the SWMM/Land model has strong potential to better inform decision makers by simulating different climate projections or flood-related strategies and evaluating the outcomes of such scenarios.

**Author Contributions:** Conceptualization, J.B., F.S., R.N. and F.F.; methodology, software, and investigation, J.B. and F.S.; supervision, R.N., F.F. and J.S.M.; visualization and writing—original draft preparation, J.B.; writing—review and editing, F.S., R.N., F.F. and J.S.M. All authors have read and agreed to the published version of the manuscript.

**Funding:** J.B. thanks Fundação para a Ciência e Tecnologia (FCT, Portugal) for providing funding through the PhD grant SFRH/BD/141880/2018.

**Institutional Review Board Statement:** Not applicable.

**Informed Consent Statement:** Not applicable.

**Data Availability Statement:** The results data presented in this work are available on request from the corresponding author if no sensible data are involved regarding the real case study.

**Acknowledgments:** The authors acknowledge Câmara Municipal de Albufeira for supplying the data required for the Albufeira case study.

**Conflicts of Interest:** The authors declare no conflict of interest.

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
