# Peer review of "Development of a 1D/2D Urban Flood Model Using the Open-Source Models SWMM and MOHID Land"

_sustainability, doi:10.3390/su15010707_

Round 1

Reviewer 1 Report

The paper covers a very interesting and relevant topic of 1D/2D coupled drainage network / surface flooding simulation. The focus on open-source software is highly relevant for the community and I am sure that the community is interested in the approach. I hope the authors can share their changes to the code (when required) and share the coupled model and an example with a link e.g. on github, that is included in the paper. However I have several major concerns.

I have questions / concerns regarding the coupling strategy. Typically 1D/2D models are directly coupled and run simultaneously (often with difficulties caused by simulation timesteps and numerical instabilities). But then the 2 models communicate in each timestep. As both models used are open-source I would have expected that such a solution is possible. The authors should please explain the ratio behind der solution and also discuss simulation times of the offline coupling approach which requires iterations model runs. I expect that for larger areas and larges flood surfaces, this approach could become extremely slow, even more as the data exchange happens trough ascii files.

For the urban drainage community please explain a bit more the capabilities of MOHID. SWMM includes several functionalities such as Low impact developments (green roof, pervious pavements, infiltration swales etc.), which are implemented in the SWMM catchment. As here the MODID rainfall runoff simulation is used, these techniques are not accessible any more. Further a paragraph on similarities and differences of MODIS and SWMM rainfall runoff models would interesting as this is relevant when existing SWMM models are changed to 1D/2D models. Further,

For the real case study it would be interesting to discuss plausibility of model results further e.g. to compare with historic flooding events when available or at least with known weak points in the system. Is it realistic that a 2year return period already results in water depths around 20-30 cm and that here are water depths of 1,5 m for return period 10. Then historical flooding date should be available. It also would be interesting to see more details on the flood event e.g. flooded areas, flooded volume, number of flooded inlet (also related to my question regarding computational performance) And also simulation time and number of required iterations should be mentioned.

Author Response

The authors wish to express gratitude to the reviewers for their time and effort in reviewing our manuscript.

Please, see the attachment with the answers to your comments on the work.

Reviewer 2 Report

The purpose of this manuscript is to evaluate the efficacy of a 1D/2D coupled drainage model to simulate urban pluvial floods in two different environments: a “synthetic street” and a small real case.

The issue is quite relevant for the journal and the scientific community.

The paper is well written and clear but the hydraulics of inlets has not been discussed in detail. In my opinion this is relevant to understand the goodness of the proposed model.

In the separate file, specific comments have been provided.

General comments

The purpose of this manuscript is to evaluate the efficacy of a 1D/2D coupled drainage model to simulate urban pluvial floods in two different environments: a “synthetic street” and a small real case.

The issue is quite relevant for the journal and the scientific community.

The paper is well written and clear but the hydraulics of inlets has not been discussed in detail. In my opinion this is relevant to understand the goodness of the proposed model.

Following specific comments are provided.

Specific comments

-          The authors often refer to urban floods but they should be more precise: they are analyzing sewer floods or urban pluvial floods. Urban floods can be produced also by storm surge of river floods. I suggest to be more precise and modify the abstract and the text of the manuscript.

-          Authors cite a work on inlet interception curves obtained from experimental studies (line 101 and Reference 22). This reference can be accompanied by another one (from the same authors) that update it and enlarge the scope to other kinds of inlets and methodologies to be used in this field. 

o   Russo B., Gómez M., Tellez J. (2021). The Relevance of Grated Inlets within Surface Drainage Systems in the Field of Urban Flood Resilience. A Review of Several Experimental and Numerical Simulation Approaches. Sustainability. MDPI. Vol. 13, No. 13, 7189. https://doi.org/10.3390/su13137189.

-          Surcharge weir and orifice equations can be used to define not only manholes’ overflow as mentioned by the authors, but also inlets overflow due to sewer pressurized conditions.

o   Gómez M., Russo B., Tellez J. (2019). Experimental investigation to estimate the discharge coefficient of a grate inlet under surcharge conditions. Urban Water Journal. Taylor & Francis. Vol. 16, No. 2, 85-91. https://doi.org/10.1080/1573062X.2019.1634107.

o   Lopes, P., J. Leandro, R. F. Carvalho, P. Páscoa, and R. Martins. 2015. “Numerical and Experimental Investigation of a Gully under Surcharge Conditions.” Urban Water Journal 12 (6): 468–476. doi:10.1080/ 1573062X.2013.831916.

-          Section 2.2.1 Coupling rational Point 3: the authors state that “when the captured flow surpasses the carrying capacity of the urban drainage infrastructures, such that the water depth inside the network reaches the surface level, part of the captured water can return to the surface through manholes’ overflow”. This is partially true because overflows could also occur by inlets (generally located at the street curb at a minor level respect to manholes usually located in the middle of the street). This aspect could be discussed and considered in the simulations. For example, the overflow in a manhole could be considered as the sum of the hydrographs of the pressurized inlets (connected by links) or considering average flow depths of the inlets. I am not sure authors are considering inlets nodes within SWMM simulations.

-          Line 181: I suppose you should add SWMM in the statement. The rationale for the coupling between MOHID Land and SWMM requires one additional layer 181 of data able to capture runoff from surface, that is, inlet devices (Figure 4).

-          Lines 213-214: please rephrase A procedure to analyze de need for successive runs is also proposed.

-          Lines 229-231. The authors do not provide information about the followed approaches or values to be used for orifice surcharged equations. These are two important items to be discussed and well explained in the text. For example, it is not clear in the two tests the characterization of inlet connections (link, weirs, cofficients used, etc.).

-          Table 3: the authors should define meaning of symbols (W: width? T: transversal slope?, L is used to times for Length and Longitudinal slopes, unit of Manning coefficient, etc.)

-          In terms of results of the test 1 (synthetic), I suggest to provide also water velocity mapping (Figure 9 b) and a short discussion about it. I suggest the authors to provide specific information and results about inlet hydraulic mechanisms.

-          Lines 300-301: The first data set is 300 used to defined different manning coefficients…

-          Lines 316: Three return periods were considered: 2 and 10 years (three or two?)

-          Results of the test 2 are not treated in detail. Again, it is not clear which equations / coefficients have been used. Also results concerning the functioning of specific inlets could be useful to understand if the model simulation has been properly carried out.

Author Response

(The authors gave the same response as above.)

Reviewer 3 Report

The paper deals with the integration of two models for urban flood simulations, namely SWMM for the drainage system (1D) and MOHID Land for the surface (2D). The coupling is “offline”, thanks to an iterative procedure. The authors show two applications: first, a simple schematic test case; then, a real test case for a Portuguese town. Overall, the paper is well written and clear, and the problem is contextualized appropriately in the introduction. The methods and results sections are also well organized.

Therefore, my suggestion is to accept the paper after a minor revision.

I suggest the authors to elaborate a little about the novelty/difference of their approach compared to existing 1D/2D models. Some hints are provided here and there along the paper (e.g., open-source codes, minimal code modification, etc.), but I think that the novelty should be stated clearly in the introduction (somewhere around lines 107-108).

Moreover, in the conclusion there is a brief discussion about the advantages/disadvantages of using an offline coupling procedure (lines 367-370). In my opinion, the authors should also discuss another disadvantage, that is the potential increase of computational times due to the necessity of repeating the simulation iteratively. This may be an issue for large domains. For example, the authors may mention how many iterations were required for convergence for the real test case and draw some observations.

Finally, I spotted a couple of typos:

-        Line 214: de -> the.

-        Line 316: three -> two (only 2 and 10 years were considered, right?).

-        Line 358: has -> as.

Author Response

(The authors gave the same response as above.)

Round 2

Reviewer 1 Report

all my comments have been addressed

Reviewer 2 Report

The manuscript has been improved and all my suggestions(comments have been adequatelly addressed. I consider the paper worth of publication in its current form.